# Length Polymorphism and Methylation Status of UPS29 Minisatellite of the *ACAP3* Gene as Molecular Biomarker of Epilepsy. Sex Differences in Seizure Types and Symptoms

**DOI:** 10.3390/ijms21239206

**Published:** 2020-12-02

**Authors:** Irina O. Suchkova, Elena V. Borisova, Eugene L. Patkin

**Affiliations:** 1Laboratory of Molecular Cytogenetics of Mammalian Development, Department of Molecular Genetics, Institute of Experimental Medicine of the Russian Academy of Sciences, St. Petersburg 197376, Russia; irsuchkova@mail.ru; 2Department of Neurology, Clinic of Institute of Experimental Medicine, St. Petersburg 197376, Russia; doc_lena@mail.ru

**Keywords:** tandem repeats length polymorphism, UPS29, ACAP3, CENTB5, centaurin beta 5, Arf GAPs, DNA methylation, epilepsy, association analysis, methyl-sensitive PCR, biomarker

## Abstract

Epilepsy is a neurological disease with different clinical forms and inter-individuals heterogeneity, which may be associated with genetic and/or epigenetic polymorphisms of tandem-repeated noncoding DNA. These polymorphisms may serve as predictive biomarkers of various forms of epilepsy. ACAP3 is the protein regulating morphogenesis of neurons and neuronal migration and is an integral component of important signaling pathways. This study aimed to carry out an association analysis of the length polymorphism and DNA methylation of the UPS29 minisatellite of the *ACAP3* gene in patients with epilepsy. We revealed an association of short UPS29 alleles with increased risk of development of symptomatic and cryptogenic epilepsy in women, and also with cerebrovascular pathologies, structural changes in the brain, neurological status, and the clinical pattern of seizures in both women and men. The increase of frequency of hypomethylated UPS29 alleles in men with symptomatic epilepsy, and in women with both symptomatic and cryptogenic epilepsy was observed. For patients with hypomethylated UPS29 alleles, we also observed structural changes in the brain, neurological status, and the clinical pattern of seizures. These associations had sex-specific nature similar to a genetic association. In contrast with length polymorphism epigenetic changes affected predominantly the long UPS29 allele. We suppose that genetic and epigenetic alterations UPS29 can modify *ACAP3* expression and thereby affect the development and clinical course of epilepsy.

## 1. Introduction

Epilepsy is a chronic neurological disorder characterized by abnormal brain activity connected with different changes at the cellular and/or synaptic levels (in particular defective arborization of the neurons, synaptic reorganization, and, in critical cases, neural cell death) [1,2,3]. The etiology of different forms of epilepsy is complex and involves genetic, epigenetic, and environmental factors [4,5,6]. In the last few years over 500 genetic loci associated with various forms of epilepsies were identified. Epilepsy-associated genes encode for ion channel and synaptic proteins, cell adhesion molecules, signaling proteins, and transcription factors. However, possible mutations in such different genes that can lead to epileptic seizures in most cases remain unclear [7,8,9]. The role of epigenetic modifications has been actively studied recently in the development of epilepsy [10,11,12,13,14].

Despite the great amount of data obtained to date on the genetics and epigenetics of epilepsy, the factors that determine the occurrence and pattern of the course of seizures are not yet identified for most forms of epileptic states. It is of particular interest that under the same external influences (for example, traumatic brain injury and neuroinfections), epilepsy does not develop in all patients. Various epileptic disorders exhibit strong heterogeneity between individuals in the pattern (severity) of disease course, in the expressivity of seizures, and the response to therapy [15]. This suggests an important role of specific genetic and/or epigenetic markers provoking or contributing to the seizure development. It is assumed that in some cases such heterogeneity may be associated with genetic and/or epigenetic (in particular DNA methylation) polymorphisms of tandem-repeated noncoding DNA minisatellite (variable number tandem repeat (VNTR)). Thus, the associations of the expanded dodecamer repeat from the cystatin B (*CSTB*) gene promoter with progressive myoclonus epilepsy (EPM1) [16,17,18] were identified. For instance, length polymorphisms of 5-HTTLPR and 5-HTTVNTR (VNTR-2, STin2) of the serotonin transporter gene (*5-HTT*) with susceptibility to temporal lobe epilepsy (TLE) [19,20], the monoamine oxidase A promoter variable number of tandem repeat (MAOA-uVNTR) with temporal lobe epilepsy with hippocampal sclerosis (TLE-HS) [21], and expansion of intronic pentanucleotide repeats of *SAMD12*, *STARD7*, *YEATS2* and *MARCH 6* genes with adult benign familial myoclonic epilepsy (BAFME) [22]. Those associations of tandem repeats with diseases may be explained by their participation in the regulation of gene expression [23,24,25]. It is known that minisatellites play an important role in the modulation of transcription, affecting efficiency mRNA translation or their stability, and can participate in alternative splicing and genome imprinting [26]. However, in most cases, the following questions remain: (a) whether the disease resulted from the only change in the number of tandem repeats or the revealed associations have also epigenetic component; (b) do the epigenetic modifications only affect repeats stability, and if they are allele-specific; and (c) if the connection between the minisatellite DNA length polymorphism and epigenetic labels exist. Thus, it is reasonable to look for new genes and genetic and epigenetic markers associated with various forms of epilepsy, in particular tandem repeats located inside or in proximity to candidate genes that are involved in the development and the functioning of the nervous system.

Human UPS29 minisatellite (46-nt repeat, 745-nt total length, Appendix A) was identified in chromosome 1p36.33 region (GenBank: AL096805.1) where the hypervariable CEB15 minisatellite is located (Gene ID: 110599576) (Figure 1). CEB15 (D1S172) is GC-rich (69.6%) VNTR (18-nt repeat, 2054-nt length) which is located in the intron of the sodium channel epithelial 1 delta subunit gene (*SCNN1D*). It was found that this minisatellite is meiotically unstable in germline cells with a paternal bias in the mutation rate [26,27,28]. Since UPS29 is also GC-rich (72.5%) (Appendix A) and is located at the distance of 6810-nt from CEB15, it was suggested that UPS29 could be unstable similarly to CEB15, and could have homologs in other mammals, in particular in rodents [29].

At present, no UPS29 homologs have been found in other mammals except for *Hominoidea* (*Pongidae family*). Only gorilla (*Gorilla gorilla*) and chimpanzee (*Pan troglodytes, Pan paniscus*) have five repeated units (total length 207-nt, 206-nt, and 255-nt, respectively) homologous (about 95%) to region UPS29 located closer to the *SCNN1D* gene (Appendix A) [32].

In silico analysis showed [31] that 46-nt imperfect repeats of UPS29 form three repeats of 226 nt. Each 46-nt repeat contains one inverted repeat (GGGGATGGCCCC), in four repeated units there is an inverted repeat (GTGCCTGTGCAC), which theoretically can form secondary DNA structures (hairpins) and cause replication slippage. Almost every repeat of UPS29 contains microsatellite (CA)_4/5_ that can act as a recombination signal and can also cause replicative slippage and non-homologous pairing during the repair of double-stranded DNA breaks. UPS29 contains also 31 imperfect repeats (GTGCCC) and has 11 sites homologous (75%) to the recombination Chi-site (CCACCACG) of *E. coli* [33]. Besides, this minisatellite has a strand asymmetry in the G:C composition, which can contribute to the formation of the Z-form of DNA and lead to unequal recombination [34]. UPS29 has six Topo II recognition sites [35,36], two potential regions of DNA fracture (GTGCACACACACGTG), and several sites (GTGCACACACACACGGTG) with high MAR potential (0.8–1.0), which theoretically can bind to the dynamic ribonucleoprotein matrix of the nucleus [37]. Thus, the results of in silico analysis indicate a potential instability of this minisatellite (Appendix A).

Our previous studies showed [31] that in the Russian population (East Slavic ethnic group, the sample of individuals from 17 to 90 years old) UPS29 was represented by seven alleles, which differed not only in the number of repeats (6, 8, 9, 10, 14, 17, and 24 ones) but also in combination of repeated units, as well as in the presence of deletions and/or single-nucleotide polymorphisms (SNPs) in DNA flanking minisatellite (Figure 1 and Appendix A, Appendix A). In the studied cohort, the allele of 17 repeats was predominant (91.5%), but the frequencies of other alleles were lower (from 0.3% to 4.4%). UPS29 heterozygosity was 12.3% according to minisatellite length estimation. Therefore, this tandem repeat was assigned to the class of low polymorphic non-hypervariable minisatellites, according to the classification of F. Denoeud with colleagues [38]. It should be noted that according to our preliminary data, there are ethnic differences in the frequency of UPS29 alleles. Short alleles (6, 8, and 9 repeats) prevail among the Mongoloid race population, in contrast to the Europeoid race. For example, the frequencies of short alleles are 89.7% among Vietnamese (Vietnam), 55.2% for Kazakhs from the east Ural, 36.0% for Bashkirs from the east Ural, and 18.3% for Bashkirs from the west Ural, but 8.5% for Russians from Northwest region of the Russian Federation [39]. Ethnic differences were described for other minisatellite repeats [40,41,42,43]. Thus, this feature of UPS29 must be considered when conducting “case-control” studies.

UPS29 was located in the *ACAP3* gene (ArfGAP with coiled-coil, ankyrin repeat, and pleckstrin homology (PH) domains 3) (GeneID: 116983) (Figure 1) [44]. To date, 13 variants of *ACAP3* transcripts were identified [30], of which only 2 were protein-coding, one -“nonsense-mediated decay”, two -“processed transcript”, and 8 -“retained intron”. However, only six transcripts have UPS29 in their introns. These are introns 14–15 (in protein-coding ACAP3–202 variant), 12–13 (in protein-coding ACAP3–201 variant), 13–14 (ACAP3–212), 5–6 (ACAP3–209), 4–5 (ACAP3–206), and 3–4 (ACAP3–204). Data on *ACAP3* expression profiles in different human organs and tissues are rather contradictory. However, its expression in the brain (both in the fetus and adults) was noted in all databases [45]. Moreover, the highest level of *ACAP3* mRNAs was shown in the cerebellum, and this level was about two times lower (relative to cerebellum level) in basal ganglia, cerebral cortex, hippocampus, amygdala, and olfactory region (Appendix A) [46,47].

*ACAP3* gene encodes protein centaurin *β5* (UniProtKB Q96P50) [48], which is a member of a multigene family of cellular proteins ArfGAPs (Ensembl Family ID: PTHR23180_SF230) (ACAP subfamily). Centaurin *β’s* protein family characterizes BAR (Bin/Amphiphysin/Rvs) domain, PH domain, zinc-finger of the phosphoinositide-dependent ArfGAP domain, and at least three ankyrin repeat (ANK) domains [49,50,51] (Appendix A). It was revealed that ACAP3 is located in the nucleoplasm and Golgi apparatus (Appendix A) [46]. ACAP3 (centaurin *β5*) is the metal-binding (Zinc), GTPase-activating protein specific to the small GTPase Arf (ADP-ribosylation factor), Arf6 with unknown physiological function. This protein is thought to be involved in the regulation of Arf6-dependent actin cytoskeleton remodeling, endocytosis (including autophagy), membrane trafficking events, cellular signaling pathway, cell adhesion, dendritic differentiation, and cell movement involving tyrosine kinase receptor [50,51,52,53]. ACAP3 is involved in neuronal migration in the cortical layer in the developing cerebral cortex, regulated morphogenesis of neurons, and neurite outgrowth through its GAP activity specific to Arf6 in mouse hippocampal neurons [54].

Thus, taking into account the structural and functional features of proteins of the Arf GAPs family, as well as their possible combined effects with other factors and/or comorbidities, ACAP3 dysfunction can be associated with human pathologies [52,55], in particular with neurological diseases. Several facts support this assumption. For example, a disturbance of the level of expression of centaurins α и γ was revealed with autism and Alzheimer’s disease [56,57]. Brain hyperexcitability in autosomal-recessive idiopathic epilepsy was connected with cytoarchitectural alterations mediated via ArfGAP6 [58]. Moreover, it should be noted that *ACAP3* belongs to the same linkage group with genes that function in the nervous system and which are associated with Parkinson’s disease, Alzheimer’s disease, mental retardation, neuroblastoma, myoclonus-like dystonia, spinocerebral ataxia with myoclonic seizures, and epilepsy (Appendix A) [59,60].

Tandem repeats (including intra-intronic ones) may be possible modulators of gene expression [23,24,25,61,62,63]. Our previous analysis revealed that UPS29 had the potential binding sites with some transcription factors such as n-Myc, BTEB, C/EBP, and CREB, which are expressed in the brain, and are involved in different cellular responses (Appendix A) [64]. Since DNA bending plays a key role in transcriptional regulation and nucleosome assembly, we also analyzed potential DNA bending within UPS29 using the BEND.it software (www.icgeb.trieste.it/dna/bend_it.html) [64]. It was revealed, that in the middle of every 46-nt repeats of UPS29 there is a GGGCC:GGCCC motif, which has been shown to direct DNA bending in vitro. Within UPS29, DNA bend value lies in the range of 0.2–2.3, whereas outside the repeat region it varies from 0.5 to 5.5. These data suggest that the structure of UPS29 may influence nucleosome assembly, and, as a consequence, the formation of transcriptionally active or inactive chromatin. Hence, UPS29 can also influence neighboring genes and even rather remote ones, as has been shown for other tandem repeats [65,66].

Taking the above data into consideration, we suggest that UPS29 may be a molecular biomarker for some human diseases, particularly neurological ones. In this connection, the present study aimed to carry out an analysis of associations of the possible participation of the *ACAP3* gene intra-intronic minisatellite length polymorphism and its methylation status in neurological diseases, in particular epilepsy.

## 2. Results

### 2.1. Association of UPS29 Length Polymorphism with Symptomatic and Cryptogenic Epilepsy (Epilepsy at the Onset, Etiology, Clinical Course)

To investigate if UPS29 could be an epilepsy risk factor, we conducted a case-control study. In all studied samples, the frequency of UPS29 genotypes corresponded to the Hardy–Weinberg equilibrium distribution. In the studied samples, we did not identify carriers of alleles from consisted of 24, 14, and 10 repeats which were revealed in our previous studied cohort [31]. Within these cohorts, there were no statistically significant differences between men and women in the frequency of short UPS29 alleles and the frequency of short allele carriers. Long UPS29 allele of 17 repeats and homozygotes 17/17 prevailed (Table 1 and Table 2). We found a statistically significant increase in the frequency of short UPS29 alleles and the frequency of individuals with short alleles among women with epilepsy compared with control (Table 1 and Table 2). This data indicates that the short UPS29 alleles increase the risk of symptomatic and cryptogenic epilepsy in women (2.5 and 3.5 times, respectively) but not in men. Here we describe a case of the association of short UPS29 alleles with epilepsy without taking into account specific etiological factors and clinical characteristics of seizures.

It should be noted that there were no significant differences between men and women in the frequency of symptomatic and cryptogenic epilepsy. In the Russian population, the latter was diagnosed 4.5 times less often than symptomatic epilepsy: In men 17.1% cases (95% CI: 10.5–26.6) and in women 18.1% cases (95% CI: 12.4–25.7). However, among subjects with short UPS29 alleles, the frequency of cryptogenic epilepsy was slightly increased (men 20.0%, 95% CI: 8.1–41.6, women 24.4%, 95% CI: 13.8–39.3) than in homozygotes 17/17 (men 16.1%, 95% CI: 9.0–26.2, women 15.1%, 95% CI: 9.1–24.2). This fact may also indicate the contribution of short UPS29 alleles to the development of cryptogenic epilepsy.

The sample number for association analysis is indeed too small for some cases here. For example, in Table 1 total allele 136, but the number of 6 repeats is only 2, which is small. Indeed, at first glance, the sample is rather small, but it is enough for nonparametric statistical criteria. Our earlier data [31,59] and later studies (unpublished data) on 450 healthy volunteers showed that short alleles frequency for the Caucasian population, is low and the size of the sample does not influence observed allele frequency. Moreover, there is a difference in the total number of alleles between men and women. It is explained by the fact that more women with epilepsy get to the clinic than men. Artificial aligning samples will lead to technical distortion of the real frequency of UPS 29 alleles in patients. Besides, used statistical criteria permit to compare samples of different sizes. It is necessary to mention that we have met long alleles of just one type (17 repeats), though, in theory, we could incorporate 24 or 10 repeat alleles. In this work in studied samples (below 50 years old), there were no alleles consisted of 24, 14, and 10 repeats found. It is necessary to note that such alleles were found by us earlier in a group of patients over 70 years old.

Since we found the association of the length polymorphism of UPS29 with epilepsy, the next step was to test the hypothesis of a possible influence of this minisatellite on the age of disease onset and the type of seizure at its onset. It turned out that the age of onset of seizure did not depend on the length of UPS29 alleles (Table 3), whereas the type of seizure at disease onset was associated with UPS29 (Table 4).

In the studied cohort of patients with epilepsy, it was found that the epilepsy debut began mainly with generalized seizures (about 70% cases, which in 2.3 times more often than focal seizures) in both men and women, in both symptomatic and cryptogenic epilepsy (Table 5). However, we revealed a decrease in the frequency of epilepsy onset with generalized seizures (55.0%, 95% CI: 34.2–74.2, n = 11, N = 20) and an increase of the frequency of onset with focal seizures (45.0%, 95% CI: 25.8–65.8, n = 9, N = 20) in men carrying short UPS29 alleles. At the same time in homozygotes for long UPS29 allele (17/17), the epilepsy onset with generalized seizures was observed in 75.8% (95% CI: 63.9–84.8, n = 47, N = 62), and with the focal ones in 24.2% (95% CI: 15.2–36.2, n = 15, N = 62) cases. It should be noted that in men with the focal seizures at the onset, the frequency of subjects with short UPS29 alleles was 2.9 times higher than in the control sample, while the frequency of short allele carriers did not differ from the control among patients with the generalized seizures at the onset. This was observed only at symptomatic epilepsy. In subjects with cryptogenic epilepsy, the frequency of short allele carriers did not differ from the control in cases of disease onset with both generalized and focal seizures (Table 4).

Unlike men, in women, the epilepsy onset started with generalized seizures in both homozygotes 17/17 and subject with short UPS29 alleles, the frequency of cases was 65.1% (95% CI: 54.6–74.4, n = 56, N = 86) and 78.1% (95% CI: 63.3–88.0, n = 32, N = 41), respectively. While the debut with focal seizures was 34.9% (95% CI: 25.7–45.4, n = 30, N = 86) and 21.9% (95% CI: 12.0–36.7, n = 9, N = 41), respectively. Moreover, in women with the epilepsy onset with generalized seizures, the frequency of carriers of short UPS29 alleles was 2.7 times (at symptomatic epilepsy) and 3.7 times (at cryptogenic epilepsy) higher than among conditionally healthy subjects, while in the case of focal seizures at the onset, the frequency of such individuals was not statistically significant compared with control (Table 4).

In the investigated group of patients with symptomatic epilepsy, the main etiological factors were perinatal pathology and traumatic brain injuries. Cerebrovascular pathologies, neuroinfection, and intoxication were observed less common, and brain tumors were revealed to be even rarer. No significant differences were found between men and women (*p* = 0.254) (Table 6). However, it should be noted that, we found a statistically significant increase of the frequency of subjects with short UPS29 alleles about four times higher (compared with the control) in men (*p* = 0.012; *phi* = +0.31; risk ratio (RR) = 4.4, 95% CI: 1.9–10.1; odds ratio (OR) = 8.9, 95% CI: 1.8–44.7) and women (*p* = 0.013; *phi* = +0.24; RR = 3.6, 95% CI: 1.6–7.9; OR = 5.7, 95% CI: 1.6–20.9) with cerebrovascular pathologies, as well as almost three-fold increase in women with traumatic head injuries (*p* = 0.008; *phi* = +0.23; RR = 2.8, 95% CI: 1.4–5.6; OR = 3.9, 95% CI: 1.5–10.1), and two-fold increase in women with perinatal pathology (*p* = 0.034; *phi* = +0.16; RR = 2.0, 95% CI: 1.1–3.8; OR = 2.4, 95% CI: 1.1–5.2). However, there were no differences with control in the cases of neuroinfections, intoxications, and brain tumors (Figure 2A).

Magnetic resonance imaging and computed tomographic scan of the brain showed hydrocephalus of various origins (including arachnoid cysts) and cortical atrophy in both men and women with epilepsy. However, brain tumors were revealed approximately two times more often in women than men (*p* = 0.029) (Table 6). It is necessary to emphasize that the frequency of carriers of short UPS29 alleles was almost four times higher in patients with cortical atrophy compared with the control samples for both men (*p* = 0.015; *phi* = +0.32; RR = 3.5, 95% CI: 1.7–7.0; OR = 5.6, 95% CI: 2.0–15.7) and women (*p* < 0.0001; *phi* = +0.37; RR = 4.1, 95% CI: 2.3–7.3; OR = 7.4, 95% CI: 3.0–18.4). While statistically significant differences of UPS29 genotypic frequencies were not observed in patients with hydrocephalus and brain tumors (Figure 2B).

In the studied cohort of patients with epilepsy, there were no statistically significant differences between men and women at their neurological status (*p* = 0.285) (Table 6). The statistically significant increase (more than three times) of the frequency of subjects with short UPS29 alleles was revealed in men with ataxia (*p* = 0.019; *phi* = +0.25; RR = 3.1, 95% CI: 1.4–6.9; OR = 4.4, 95% CI: 1.3–14.7), with pyramidal system disturbances (*p* = 0.006; *phi* = +0.32; RR = 4.3, 95% CI: 1.9–9.4; OR = 8.3, 95% CI: 2.0–35.5), and with sensory impairment (*p* = 0.025; *phi* = +0.29; RR = 4.6, 95% CI: 1.9–11.2; OR = 10.0, 95% CI: 1.5–66.2) compared with control. The frequency of subjects with short alleles for women was significantly higher compared to control only with epilepsy, accompanied by autonomic dysfunction (*p* = 0.001; *phi* = +0.29; RR = 3.6, 95% CI: 1.8–6.9; OR = 5.6, 95% CI: 2.0–15.6) (Figure 2C).

### 2.2. Association of UPS29 Length Polymorphism with Clinical Features of Seizure

The seizure pattern (type, frequency, duration, and the moment of seizure onset) is one of the critical factors of the clinical course of epilepsy. Based on this information, clinicians choose the appropriate treatment and prognosis of the course of the disease. Therefore, the hypothesis of a possible association of UPS29 with a seizure pattern was verified in our work.

In the studied cohort of patients with epilepsy, we did not reveal statistically significant differences between men and women on the type (*p* = 0.082), frequency (*p* = 0.617), and start time of seizures during the day (*p* = 0.995), except for the duration of the seizures (*p* = 0.028). In the latter case in women, seizures duration less than 1 min was approximately 1.4 times more likely than in men, while seizures duration 10–20 min, on the contrary, were 3 times less common than for men. Generalized seizures, seizures with a frequency of 2–4 times a month, duration seizures about 5 min, and seizures that occur in the daytime or at any time of 24 h were most often observed in both men and women (Table 7).

In the studied sample of patients with epilepsy, an increase (from 2.5 to 3 times) in the frequency of subjects with short UPS29 alleles was found among men (*p* = 0.026; *phi* = +0.23; RR = 2.9, 95% CI: 1.2–6.6; OR = 4.0, 95% CI: 1.2–13.0) and women (*p* = 0.001; *phi* = +0.29; RR = 3.3, 95% CI: 1.8–6.1; OR = 4.9, 95% CI: 2.0–11.9). There were observed focal seizures among men (*p* = 0.035; *phi* = +0.23; RR = 3.0, 95% CI: 1.2–7.06; OR = 4.1, 95% CI: 1.2–14.9). For women (*p* = 0.013; *phi* = +0.21; RR = 2.6, 95% CI: 1.3–5.3; OR = 3.4, 95% CI: 1.3–8.9) we observed patients with combined seizures (focal and generalized). Only in women did we observe generalized seizures (*p* = 0.003; *phi* = +0.22; RR = 2.3, 95% CI: 1.4–4.3; OR = 3.1, 95% CI: 1.5–6.4) as compared to healthy individuals (Figure 3A). At the same time in men, we did not reveal statistically significant differences in the frequency of individuals with short UPS29 alleles among subgroup of patients with epilepsy in such characteristics of seizures as frequency and duration (compared with the control, *p* > 0.05). In women, the frequency of carriers of short UPS29 alleles was 3 times higher than in the control, among patients with daily seizures (*p* = 0.005; *phi* = +0.26; RR = 3.4, 95% CI: 1.7–7.0; OR = 5.3, 95% CI: 1.8–16.3) and seizures occurring 2–4 times per month (*p* < 0.0001; *phi* = +0.32; RR = 3.2, 95% CI: 1.9–5.4; OR = 4.8, 95% CI: 2.4–9.5) (Figure 3B). Carriers of short UPS29 alleles were 4 times more often among women with seizures duration less than 1 min (*p* < 0.0001; *phi* = +0.46; RR = 4.8, 95% CI: 2.8–8.1; OR = 10.6, 95% CI: 4.5–25.1) and 5 min (*p* = 0.007; *phi* = +0.20; RR = 2.3, 95% CI: 1.3–4.1; OR = 2.8, 95% CI: 1.4–5.8) (Figure 3C). An association of short UPS29 alleles was also revealed for the time of day of the onset of seizures. Compared with the control, individuals with short UPS29 alleles were approximately 3 times more often among patients whose seizures occurred during the daytime both in men (*p* = 0.003; *phi* = +0.28; RR = 2.9, 95% CI: 1.5–5.8; OR = 4.1, 95% CI: 1.6–10.4), and women (*p* < 0.001; *phi* = +0.28; RR = 3.0, 95% CI: 1.7–5.2; OR = 4.1, 95% CI: 2.0–8.7), as well as in women with seizures at any time of day (*p* = 0.012; *phi* = +0.19; RR = 2.3, 95% CI: 1.2–4.1; OR = 2.8, 95% CI: 1.3–6.0). However, patients with seizures in the morning were not found among carriers of short UPS29 alleles (Figure 3D).

We did not find any significant association between short allele frequency and epilepsy among men. However, there is a significant association between etiological factors, structural changes, or neurological conditions among male epileptic patients with short allele frequency compared to healthy substances. This observation is rather difficult to explain. Firstly, this discrepancy just underlines the imperfection of the classification of epilepsy into symptomatic, cryptogenic, and idiopathic, and speaks in favor of the classification of epilepsy according to the type of seizures at the onset of the disease and during the disease, as well as by specific etiological factors of the disease (where this can be established during clinical examination and patient interview). Secondly, it is necessary to take into consideration that UPS29 is located in maternally expressed imprinted locus 1p36 (Figure 1, and see below). Thus, it cannot be excluded a probability that the manifestation of symptoms is determined mainly by long allele. Inevitably, this question needs additional studies.

### 2.3. Association of UPS29 Methylation Status with Epilepsy Type and Clinical Features of the Seizure

Since the identification of epigenetic mechanisms of epilepsy is a prospective study that can provide additional information on the pathophysiology and treatment of epilepsy, in this work we analyzed the DNA methylation of UPS29 minisatellite in patients with a different clinical picture of seizures. These data, in turn, may indicate a possible molecular mechanism of the UPS29 effect in the pathogenesis and course of epilepsy.

In all the studied cohorts, we did not find statistically significant (intragroup) differences between men and women in the frequency of methylated and hypomethylated forms of UPS29 for both the long allele of 17 repeats and the short alleles of this minisatellite (Table 8) and also in the frequency of UPS29 epigenotypes (Table 9). However, the frequency of individuals with hypomethylated (unmethylated) UPS29 alleles increased statistically significantly in both men and women in comparison with the control in 1.6 and 1.4 times, respectively, among patients with symptomatic epilepsy (Table 9). This result was due to an increase in the fraction of hypomethylated forms of long (17 repeats) allele, rather than short UPS29 alleles as we expected. Unlike men in women, the frequency of hypomethylated long allele was increased at cryptogenic epilepsy also (Table 8). These data indicate the possible epigenetic role of the UPS29 long allele in the pathogenesis of epilepsy. Therefore, at the next stage, we carried out the association analysis of hypomethylated UPS29 alleles with structural changes in the brain, neurological status, and clinical pattern of seizures in patients with epilepsy.

So, the association of UPS29 hypomethylation with structural changes in the brain was revealed only in men, where there was a statistically significant increase in the frequency of hypomethylated alleles (predominantly long allele) in patients with cortical atrophy (*p* = 0.031; *phi* = +0.16; RR = 2.0, 95% CI: 1.2–3.0; OR = 3.8, 95% CI: 1.1–13.3) and brain tumors (*p* = 0.015; *phi* = +0.20; RR = 2.9, 95% CI: 2.4–3.6) (two- and three-fold increase, respectively) (Figure 4A). Furthermore, only in men we revealed the association of UPS29 hypomethylation with pyramidal system disturbances (*p* < 0.001; *phi* = +0.26; RR = 2.3, 95% CI: 1.7–3.1; OR = 6.7, 95% CI: 2.1–21.3), with ataxia (*p* = 0.001; *phi* = +0.23; RR = 2.0, 95% CI: 1.4–2.7; OR = 3.8, 95% CI: 1.7–8.7), with cranial nerve impairment (*p* = 0.029; *phi* = +0.16; RR = 1.8, 95% CI: 1.2–2.6; OR = 2.9, 95% CI: 1.1–7.4), and autonomic dysfunction (*p* = 0.028; *phi* = +0.15; RR = 1.5, 95% CI: 1.1–2.1; OR = 2.1, 95% CI: 1.1–4.0), where the frequency of hypomethylated alleles was approximately 2 times higher than in the control (Figure 4B).

The association of UPS29 hypomethylation with the clinical pattern of seizures showed that for all types of seizures (focal, generalized, and combined), the frequency of hypomethylated UPS29 alleles increased in both men and women, while the highest increase (almost 3 times) was observed in women with absence (Table 10, Figure 5A). A significant increase in the frequency of hypomethylated UPS29 alleles was found among patients (in both men and women) with the daily seizures and seizures “2–4 times per month”. In the case of seizures “One time per month”, an increase in the frequency of hypomethylated alleles was shown only in men, while in women, on the contrary, there was a sharp decrease in these alleles and an increase in the frequency of methylated forms of UPS29 (Table 10, Figure 5B). The increased frequency of hypomethylated UPS29 alleles was in women with a duration of seizures of less than 1 min and about 5 min, while in men this was observed not only among patients with 5 min seizures but also with seizures of 10–20 min and varied (sec, 1, 5 or 10 min) (Table 10, Figure 5C). Compared to the control, an increase in the frequency of hypomethylated UPS29 alleles was observed in both men and women with seizures “In the daytime” and “Any time of 24 h”, but unlike women in men, such alleles were often found among patients with seizures “At night” (Table 10, Figure 5D). The homozygous hypomethylated short allele was not found. A probable explanation points to the same reason as above, namely that among Caucasians, short UPS29 alleles in a homozygous state are very rare. Therefore, in this work, we were unable to identify hypomethylated short alleles in the homozygous state.

## 3. Discussion

The obtained data indicate that UPS29 length polymorphism and its DNA methylation status are the risk factors of epilepsy and/or possible modulators of the clinical course of this disease (structural changes in the brain, neurological status, seizures pattern–type, frequency, duration, and the time of 24 h of seizure onset). At the same time, the sex-specific effect of UPS29 was noted.

We revealed the association of short UPS29 alleles with symptomatic and cryptogenic epilepsy only for women (in both homozygotes and heterozygotes). Unlike men in women, short UPS29 alleles possibly represent an increased risk of seizures among patients with perinatal pathology or after brain injury, although the comorbidity of cerebrovascular pathologies and epilepsy was observed in both women and men. Moreover, in both women and men with epilepsy, the presence of short UPS29 alleles in the genome was associated with an increased risk of developing cortical atrophy, but there were no associations with hydrocephalus and brain tumors. Among investigated patients, only in women, the short UPS29 alleles were associated with autonomic dysfunction, whereas in men this association was observed among individuals with ataxia, pyramidal system disturbances, and sensory impairment.

In patients with short UPS29 alleles, epilepsy at onset started predominantly with focal seizures in men, and with generalized seizures in women. However, then (in the course of the disease) the effect of short UPS29 alleles on the clinical pattern of seizures was changed. Thus, in men, the short UPS29 alleles were associated not only with focal seizures but with combined (focal and generalized) ones, and in women—not only with generalized seizures but with focal and combined (focal and generalized) ones.

To explain the possible molecular mechanism of the influence of short UPS29 alleles on the pathogenesis and clinical course of symptomatic and cryptogenic epilepsy, we checked the DNA methylation status of this minisatellite. Unexpectedly, the frequency of the hypomethylated UPS29 alleles was increased in patients with epilepsy (compared with the control), at that predominantly long allele of 17 repeats rather than short alleles (9, 8, and 6 repeats). At the same time, an increase in the frequency of hypomethylated UPS29 alleles was revealed not only in men but also in women when the association analysis was carried out between UPS29 methylation status and the clinical pattern of seizures (types of seizures, seizure frequency, duration, and the time of day of seizure onset). Earlier it was shown that VNTRs influenced the transcription activity of genes in which they are located (19, 20, 77). We did not observe significant differences in allelic and genotypic frequencies of 5-HTTLPR and STin2 among patients with symptomatic and cryptogenic epilepsy (cohort from the presented study) in comparison with the control sample [67], but in these patients, we revealed an association with short UPS29 alleles of *ACAP3* gene in both homozygotes and heterozygotes. Besides, a statistically significant increase in the frequency of individuals with short UPS29 alleles was also found among women with early and late-onset Parkinson’s disease [59]. Indeed, earlier there were described seizures in patients with Parkinson’s disease [68,69]. The molecular mechanism of the revealed association of UPS29 with symptomatic and cryptogenic epilepsy and Parkinson’s disease is not clear, as well as for most tandem repeats in similar studies. Sure, to explain the molecular mechanism behind this disease associated with the hypomethylated short allele of UPS29 it would be stronger if we could present the mRNA/protein level of ACAP3 in patients harboring hypomethylated short allele vs. hypermethylated long alleles. As was mentioned above, short UPS29 alleles in a homozygous state of occurrence are very rare among Caucasians, therefore it was not possible to conduct an adequate statistical analysis of the mRNA / protein level of ACAP3 expression in patients with epilepsy, in whom this allele is in a hypomethylated state in comparison with similar genotype, but in a methylated state, as well as in comparison with similar genotypes/epigenotypes in the control group in men and women.

Our previous investigations on mammalian cell cultures that were transfected with plasmid constructs contained UPS29 (allele 17 or 6) and the reporter *EGFP* gene under eukaryotic promoter ROSA26, showed that UPS29 enhanced the reporter gene expression in cells of neuronal origin. The most prominent effect was observed for the short UPS29 allele [70,71]. This data indicated that UPS29 has enhancer-like activity in cells of neuronal type. Therefore, we suggest that the structural and/or epigenetic disturbances in this minisatellite may lead to the change of the *ACAP3* gene expression (perhaps and/or neighboring genes) thereby causing pathological and/or physiological changes in the brain.

It should be noted that in peripheral blood leukocytes we found an increase of *ACAP3* mRNA level in women and men with epilepsy, and also in women with early-onset Parkinson’s disease (5, 2.5, and 6-fold, respectively), while the mRNA level of *CSTB* gene was decreased at epilepsy and Parkinson’s disease only in women (3- and 2.5-fold, respectively) in comparison with control [72]. The decreased *CSTB* expression associated with the expansion of VNTR was also described in patients with progressive myoclonus epilepsy [73]. These data indicate the possible involvement of *ACAP3* and *CSTB* genes in the pathogenesis for both symptomatic epilepsy and Parkinson’s disease, which at least partly explains the above-mentioned seizures in patients with Parkinson’s disease. 

DNA methylation is known to be one of the main epigenetic mechanisms for gene activity regulation and genome instability [74,75,76]. Tandem repeats are potential targets for this kind of modification and are prone to instability partly due to the active demethylation process [77,78,79,80]. It has already been established the role of repeats and DNA methylation in the formation of secondary DNA structures, genomic instability, and the organization of the scaffold/matrix attachment regions (S/MAR), which are epigenetic modifiers and, as a consequence, risk factors for the development of neuropathologies [84¨CC85-87]. It is known the hypervariable minisatellites of some genes interact with transcription factors [63]. Therefore, the effect of UPS29 on the expression of ACAP3 (and /or neighboring genes) may be associated not only with DNA methylation level but also with structural changes in the nucleotide sequence, for example, at putative binding sites with transcription factors (Appendix A). On the other hand, UPS29 is GC-rich, prone to secondary structures, and has high MAR potentials as was shown [64]. In silico analysis showed [64,70] that some of the internal repeats are deleted in short UPS29alleles, with which the transcription factors can interact such as SP1, TFIID, AP-1, CACCC-binding protein, as well as AP-2, BTEB, C/EBP, and CREB (specific for nervous system cells) (Appendix A). On the other hand, UPS29 is GC-rich, prone to secondary structures, and has high MAR potentials as was shown [64]. Thus, we assume that the structural and epigenetic changes in UPS29 in the context of predicted parental-of-origin ACAP3 expression can answer the questions:

Why do not all carriers of short UPS29 alleles have epilepsy and why homozygotes 17/17 with this disease occurred (Appendix A). External factors can change the level of DNA methylation and, as a result, change the length and even the composition of tandem repeats, in turn, affecting gene expression. Thus, the individual and gender differences noted by us can result in susceptibility to the development of various forms of neurological pathologies, including epilepsy. Furthermore, epigenetic factors (including DNA methylation) play a significant role in brain development and neuronal function and may contribute to the developmental neuropathologies, in particular epilepsy [11,12,13,14,74]. Indeed, we found a significant increase in the frequency of unmethylated UPS29 forms for patients with epilepsy, while there were no differences compared with controls among individuals with Parkinson’s disease [72]. It can be assumed that the increase in the level of ACAP3 expression described by us earlier [72] could be associated with a decrease of UPS29 methylation level in patients with epilepsy, taking into account the mentioned coincidence between seizures and Parkinson’s disease [68,69].

In this work, we observed clear differences between men and women both in the composition of the minisatellite UPS29 and in the level of its methylation. Sex difference in seizure types and symptoms was found earlier [81] as well as in patients with Human Cancer and Autoimmunity [82]. Possible mechanisms of such sex bias are far from clear. It can be assumed that this is due to the phenomenon of imprinting. Indeed, it was found that the imprinted genes are predominantly expressed in the brain [83,84,85], and ACAP3 is the predicted imprinted gene, and it is expressed from the maternal allele (Appendix A) [83]. Since there is evidence of the involvement of tandem repeats in the control of genomic imprinting [86,87,88], it may be assumed that UPS29 is involved in this process. Perhaps this role of the minisatellite is associated with the presence in its composition of binding sites for transcription factors USF [89] and YB-1 [90], which are involved in epigenetic mechanisms of regulation of gene expression, including genomic imprinting. Moreover, there is data showing that therapy with certain drugs [91] plays a role in tandem repeats in the sex-dependent response of the organism. Therefore, it is possible that ACAP3 as a GTPase-activating protein [92] can react to both neurotransmitters and sex hormones, and the abnormalities in ACAP3 expression, in turn, can cause the development of sex-dependent clinical characteristics of epilepsy. These features of UPS29 can affect not only the *ACAP3* gene but also neighboring *SCNN1D* [93] and even distant *ACOT7*, which is critical for the functioning of neurons and development of mesial temporal lobe epilepsy, as shown for other minisatellites [65,66]. This may also be one of the explanations for the occurrence of comorbidity of epilepsy with vascular [94,95,96], psychiatric [97], and oncological diseases [98,99]. For example, the disturbance of *SCNN1D* expression can lead not only to deviations in neuronal conductivity but also to cerebrovascular pathologies, structural changes in the brain, as well as the development of neuroblastoma [100,101], that were observed in some individuals in the investigated cohort of patients with symptomatic and cryptogenic epilepsy.

Returning to the possible mechanisms of pathogenesis of symptomatic and cryptogenic epilepsy, it is important to consider the fact that we found an association of short UPS29 alleles with structural changes in the brain, namely cortical atrophy, whereas the association with hydrocephalus and arachnoid cysts absented, although they are often associated with the development of epilepsy [102]. These data support the functional significance of UPS29 and the *ACAP3* gene in neurogenesis (neuropathogenesis) since the role of ACAP3 in neuronal migration in the developing cerebral cortex has already been proven [54]. Other proposed ACAP3 functions are cellular signaling pathway and visual traffic [103,104], the disturbances of which may be responsible for deviations in synaptic plasticity and neuronal conduction. This fact may explain the identified associations of UPS29 with the clinical characteristics of epileptic seizures (type, duration, and frequency), assuming that this tandem repeat is involved in the modulating of *ACAP3* expression.

## 4. Materials and Methods

### 4.1. Participant Recruitment (Patients and Healthy Volunteers)

This study was performed following the Declaration of Helsinki and approved by the local independent Ethical Committee of Federal State Budgetary Scientific Institution “Institute of Experimental Medicine”: Project–08–04–12167 (17 May 2012), Project–10-04-00676 (16 February 2015), Project–11–04–00254 (24 March 2016). All subjects provided written informed consent to participate in this research. The study involved 209 patients with epilepsy (82 males and 127 females) and 226 healthy unrelated volunteers (92 males and 134 females). All subjects were residents of St. Petersburg (Russian Federation), ethnic origin-Caucasian.

The cohort of patients with epilepsy consisted of individuals, who have visited the Neurologic Clinic of the Institute of Experimental Medicine from January 2008 to December 2019. The patients were diagnosed according to the 1989 Classification of Epilepsies and Epileptic Syndromes proposed by the Commission on Classification and Terminology of the International League Against Epilepsy [105]. All patients had a comprehensive diagnostic evaluation, including a detailed history (heredity, prior perinatal pathology, brain trauma, neuroinfections, brain tumors, cerebrovascular pathology, etc.) and clinical data of epilepsy (age at onset, type, frequency, and duration of seizures), neurological examination, neuropsychological testing, magnetic resonance imaging or computed tomographic scan, and surface electroencephalographic. At the beginning of this project, for genetic and epigenetic investigations this cohort was divided into two groups based on the etiology of seizures: (a) symptomatic epilepsy, associated with any event that damages the brain; and (b) cryptogenic (probably symptomatic) epilepsy-defined here as epilepsy of presumed symptomatic nature in which the cause has not been identified. The group “Symptomatic epilepsy” consisted of 68 males (mean age 36.1 ± 15.3 years) and 104 females (mean age 35.0 ± 13.7 years). The group “Cryptogenic epilepsy” included 14 males (mean age 28.0 ± 5.8 years) and 23 females (mean age 30.0 ± 11.8 years). As data accumulated and for a deeper analysis of UPS29 associations with the clinical characteristics of seizures, these cohorts were divided into subgroups taking into account the new 2017 ILAE classification of epilepsy and epileptic syndromes (the type of seizure at the onset and during the disease) [106].

The control group consisted of volunteers from among students, graduate students, and researchers of St. Petersburg University, St. Petersburg Polytechnic University of Peter the Great, St. Petersburg Medical Academy of Postgraduate Education, Pavlov Medical University, Military Medical Academy named after S. M. Kirov, and the Institute of Experimental Medicine. The volunteers underwent a rigorous clinical and neurological examination were healthy at blood drawing and did not have a history of neurological disease. In this cohort, the mean age was 35.0 ± 13.9 years in males and 32.6 ± 12.4 years in females.

### 4.2. Genome DNA Isolation and Genotyping

Peripheral venous blood samples (5 mL) incubated at 37 °C for 30 min. The formed middle phase (white blood cells) was transferred into microcentrifuge tubes and incubated in 900 μL 5 mM EDTA at room temperature for 15 min and centrifuged at 5000 rpm. Then the leukocyte pellet was lysed in buffer (100 mM NaCl, 50 mM Tris-HCl pH 8.0, 10 mM ЭДTA pH 8.0, and 1% SDS) in the presence of 0.1 mg/mL of proteinase K, overnight at 37 °C. Genome DNA was extracted from leukocytes using the chloroform-isoamyl alcohol purification protocol and the ethanol precipitation method. DNA quantity and quality were assessed by NanoDrop 2000C (Thermo Scientific, Moscow, Russia) and 1% agarose gel electrophoresis.

Subjects were genotyped for the UPS29 minisatellite length polymorphisms. The genotyping reactions were performed blinded to clinical features. Human UPS29 was amplified by polymerase chain reaction (PCR) using the following oligonucleotide primers: Forward (EcoRI-containing) 5′-gtcagaattccgcgagagccctgacagttg-3′ (*fwPR*) and reverse (HindIII-containing) 5′-tcataagcttcacatgggcagatggtacctgc-3′ (*revPR*) (these primers can be used in other studies to clone UPS29 PCR products in different plasmid vectors) (Appendix A). PCR mixture (final volume 25 μL) contained 60 mM Tris-HCl pH 8.5, 25 mM KCl, 10 mM β-mercaptoethanol, 0.1% Triton X-100, 3 mM MgCl_2_, 0.32 mM each dNTP, 1.25 U Taq-polymerase (Medigen, Novosibirsk, Russia), 0.4 μM each primer, and 10–20 ng of genomic DNA. PCR was run on a thermal cycler “TP4-PCR-01-Tertzik” (DNA-Technology, Moscow, Russia). The thermal profile was 97 °C for 7 min, followed by 30 cycles of denaturation at 96 °C for 60 s, annealing at 57 °C for 60 s, followed by extension at 72 °C for 3 min, with a final extension at 72 °C for 8 min. PCR products were resolved by neutral electrophoresis in 6% polyacrylamide gel (PAAG), and bands visualized by staining with 0.1% AgNO_3_. The amplification product size was estimated against a standard 100 bp DNA ladder (Medigen, Novosibirsk, Russia). PCR bands size for UPS29 alleles consisting of 17, 9, 8, and 6 repeat units were 900, 520, 500, and 400 bp, respectively (for 100 bp DNA ladder in 6% PAAG) [31] (Appendix A). Our classification of UPS29 alleles was based on the minisatellite length (variation in repeat unit number) without assessing possible internal variations into repeat units. In this paper, UPS29 alleles were divided into two groups depending on the minisatellite length: (1) Long allele of 17 repeat units, prevailing in healthy Russian population; and (2) Short alleles (alleles of 9, 8, and 6 repeats), which were grouped, because they were rare in this population [31]. These two groups of UPS29 alleles (according to their size) were used for case-control analysis.

### 4.3. Analysis of UPS29 DNA Methylation Status

In silico analysis of UPS29 revealed that the nucleotide sequence allele of 17 repeats contained 12 recognition sites (5′-CCGG-3′) for MspI and HpaII endonucleases, therefore UPS29 DNA methylation status was determined by methyl-sensitive PCR using these isoschizomers [107]. Three probes were made for each human DNA sample: With MspI, with HpaII, and without enzymes (only buffer). The latter served as a control for the preservation of DNA in the reaction mix. The enzymatic hydrolysis of 100 ng genome DNA was carried out by 10 U MspI or 10 U HpaII (Thermo Scientific) in 50 μL buffer (10 mM Tris-HCl pH 7.6, 10 mM MgCl_2_, 1 mM dithiothreitol) at 37 °C overnight. Then 7.5 μL these solutions (with MspI or HpaII digested, or intact genome DNA) were added in PCR mixes for following UPS29 amplification as was described above. However in this protocol, the collected PCR buffer was magnesium-free, since the required amount of these ions were contained in the adding volume of restriction mix. PCR products were resolved by neutral electrophoresis in 6% PAAG, and bands were visualized by staining with 0.1% AgNO_3_. pBR322 plasmid DNA (1000 ng per 10 U MspI or 10 U HpaII) served as a positive control of enzymatic hydrolysis efficiency. DNA integrity was evaluated from probes without enzymatic hydrolysis. The completeness of genome DNA cleavage was evaluated by MspI probes. UPS29 methylation status was evaluated by HpaII probes. The presence of PCR bands of UPS29 indicated its methylation, while the absence of these products talked about its hypomethylation/unmethylation into CCGG sites. This method does not allow to determine the number and position of methylated CpG in the target sequence DNA but enables sufficiently quick identification of DNA samples (of individuals) with methylated or hypomethylated (unmethylated) UPS29. Since this method does not allow us to identify one or both alleles that were methylated in the case of UPS29 homozygotes, we included such individuals in the group “Methylated UPS29” (as with two methylated alleles) in the statistical analysis of the data.

### 4.4. Statistical Analysis

The clinical variables and UPS29 genotypes/epigenotypes were compared between patients with epilepsy and a control group. In a genetic investigation, our cohorts were stratified into two groups according to the presence/absence of short UPS29 alleles. In an epigenetic investigation, our cohorts were stratified into two groups according to the presence/absence of hypomethylated UPS29 alleles. To verify that UPS29 genotype frequencies were in Hardy–Weinberg equilibrium was used the Chi-square analysis. Differences in the frequencies of UPS29 genotypes/epigenotypes and alleles/epialleles were performed using the contingency tables (2 by 2, 2 by 3, 2 by 4) with Chi-square-test (for large samples) or Fisher’s Exact test (for small samples). In cases of statistically significant differences between the control group and patients with epilepsy were calculated standard measures for risk ratio, odds ratio, and the Phi coefficient of association. RR and OR were calculated with a 95% confidence interval (95% CI). The age of epilepsy onset between groups (symptomatic and cryptogenic epilepsy) was compared with the Kruskal–Wallis test followed by the Dann test for pairwise comparison. Dunn-test *p*-values were adjusted by the Benjamini–Hochberg method (correction for multiple comparisons). All statistical analysis was two-tailed, and Bonferroni’s corrected *p*-values < 0.05 were considered statistically significant. 

UPS29 allele/genotype frequencies data are shown as *n* (%) and 95% CI. The age of subjects at the time of the study and the age at onset of epilepsy are indicated as the mean and standard deviation (M ± SD). 

Statistical analyses were conducted using Microsoft Excel and free on-line statistical software: http://vassarstats.net/; https://wpcalc.com/en/equilibrium-hardy-weinberg/; http://astatsa.com/KruskalWallisTest/; http://www.husdyr.kvl.dk/htm/kc/popgen/genetik/applets/kitest.htm. 

All diagrams and graphs were plotted using Microsoft Excel.

## 5. Conclusions

Our data indicate that short UPS29 allele and hypomethylation of this tandem repeat may be new genetic/epigenetic markers of some epilepsy form and seizures pattern. There was also observed the sex difference in these characteristics. Moreover, the ACAP3 gene, in which UPS29 is located, may play a role in the development of this disease and other neurological pathologies under environmental influences. The structural and epigenetic changes of intra-intronic minisatellite observed in this work upon epilepsy and perhaps for other neurological disease points to possible usage of nucleic acid (in particular non-coding tandem repeats) therapies as was suggested in [108,109,110]. All these statements concerning the molecular mechanisms of action of UPS29 and ACAP3 in the etiopathogenesis of epilepsy require further research.

## Figures and Tables

**Figure 1 ijms-21-09206-f001:**
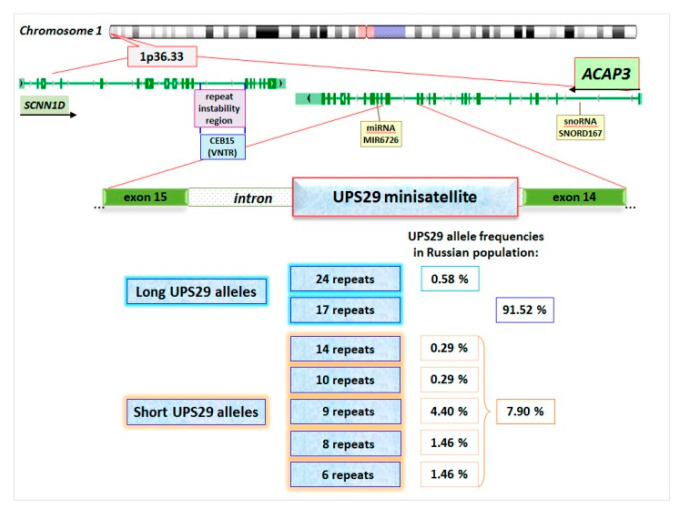
Scheme of chromosomal localization of *ACAP3* and sodium channel epithelial 1 delta subunit gene (*SCNN1D*) genes, and intragenic location of UPS29 and CEB15 minisatellites. Allelic frequencies of long and short alleles UPS29 in the Russian population [30,31].

**Figure 2 ijms-21-09206-f002:**
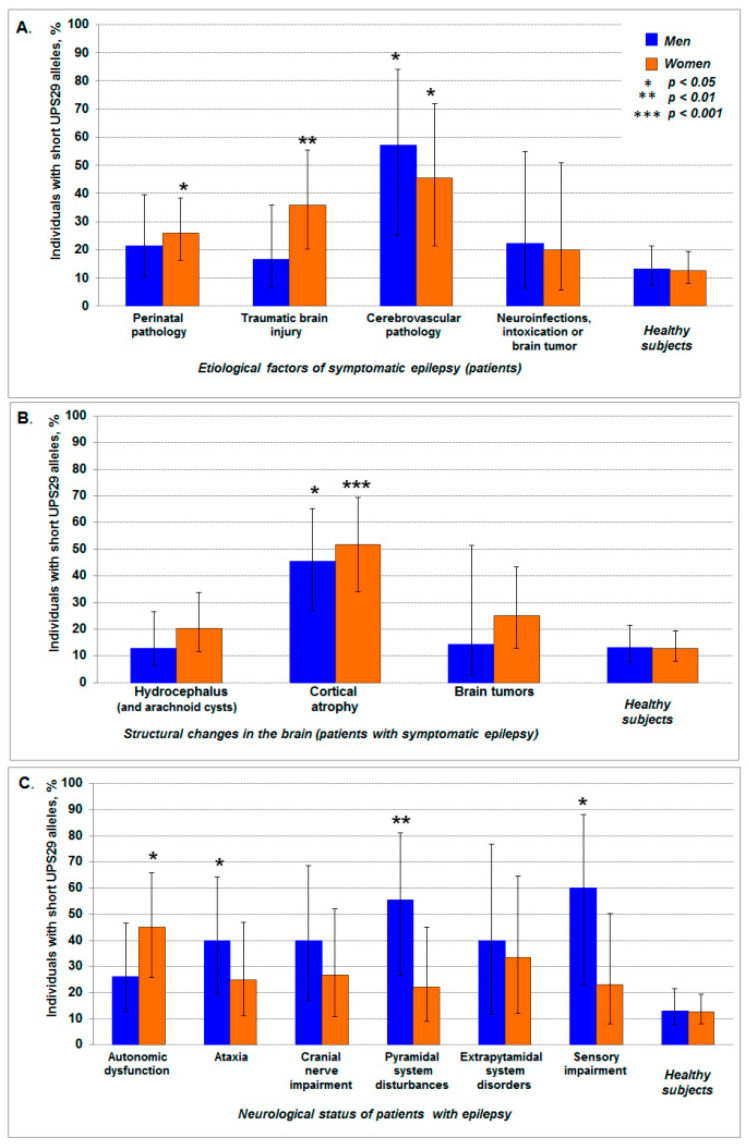
The frequency of carriers of short UPS29 alleles among patients with epilepsy of different etiology (**A**), structural changes in the brain (**B**), and neurological status (**C**). The data is shown with 95% CI. The asterisk indicates statistically significant differences in comparison with control (healthy subjects).

**Figure 3 ijms-21-09206-f003:**
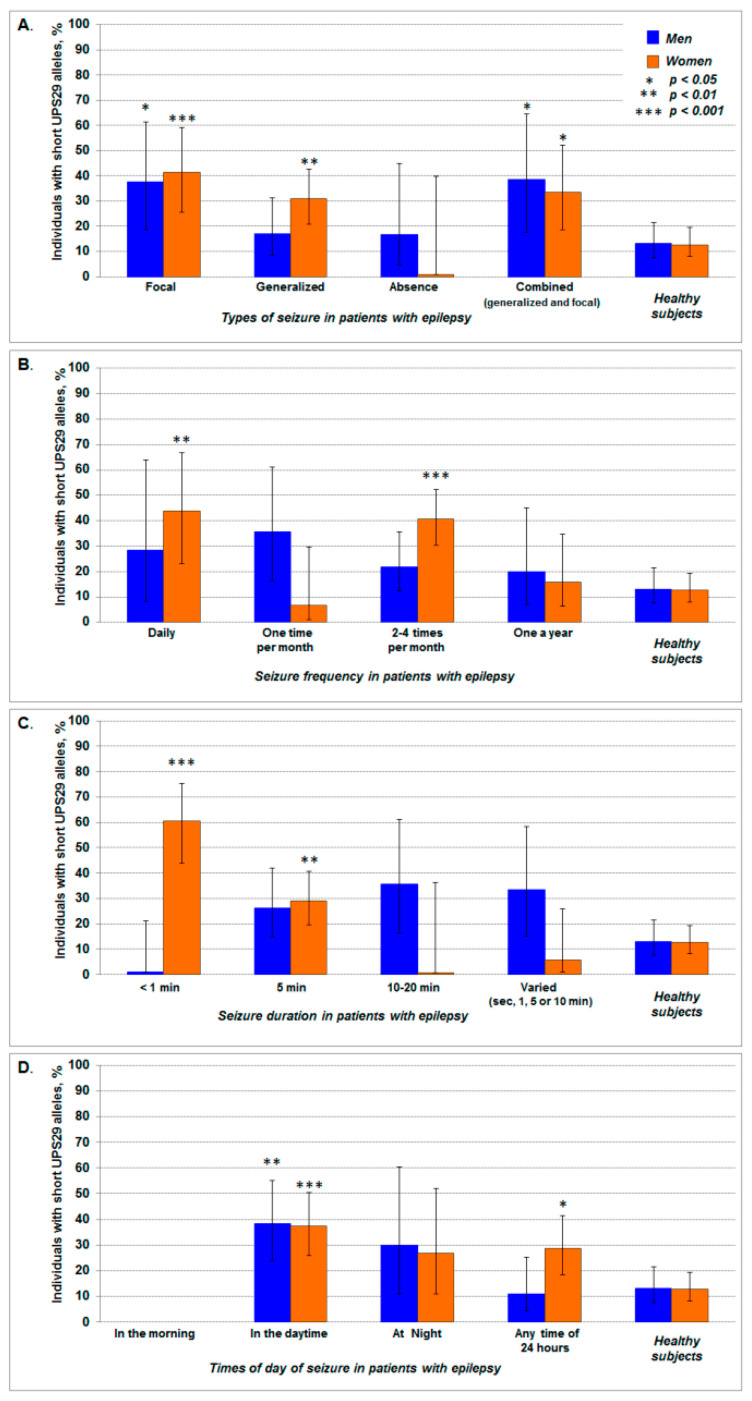
The frequency of carriers of short UPS29 alleles among patients with different clinical patterns of seizures: type (**A**), frequency (**B**), duration (**C**), and time of day of seizure onset (**D**). The data is shown with 95% CI. The asterisk indicates statistically significant differences in comparison with control (healthy subjects).

**Figure 4 ijms-21-09206-f004:**
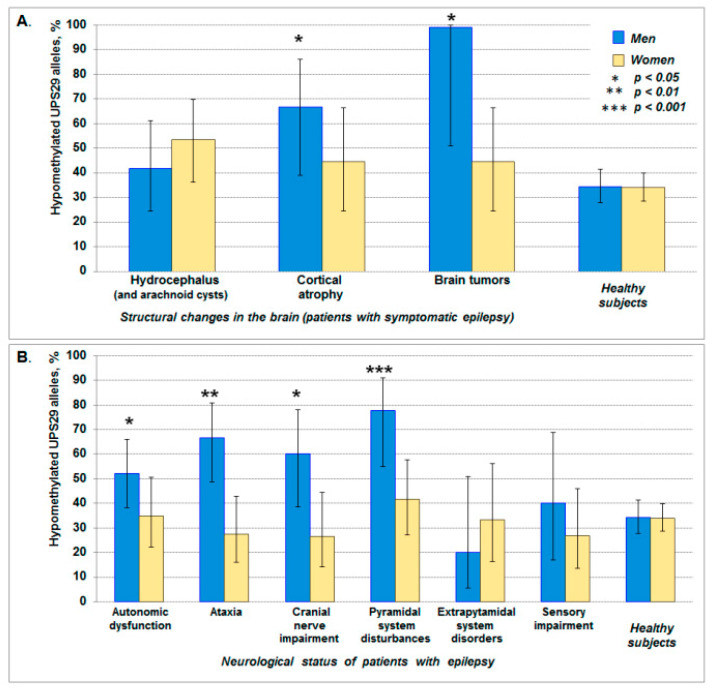
The frequency of hypomethylated (unmethylated) UPS29 alleles in patients with epilepsy with different structural changes in the brain (**A**) and neurological status (**B**). The data is shown with 95% CI. The asterisk indicates statistically significant differences in comparison with control (healthy subjects).

**Figure 5 ijms-21-09206-f005:**
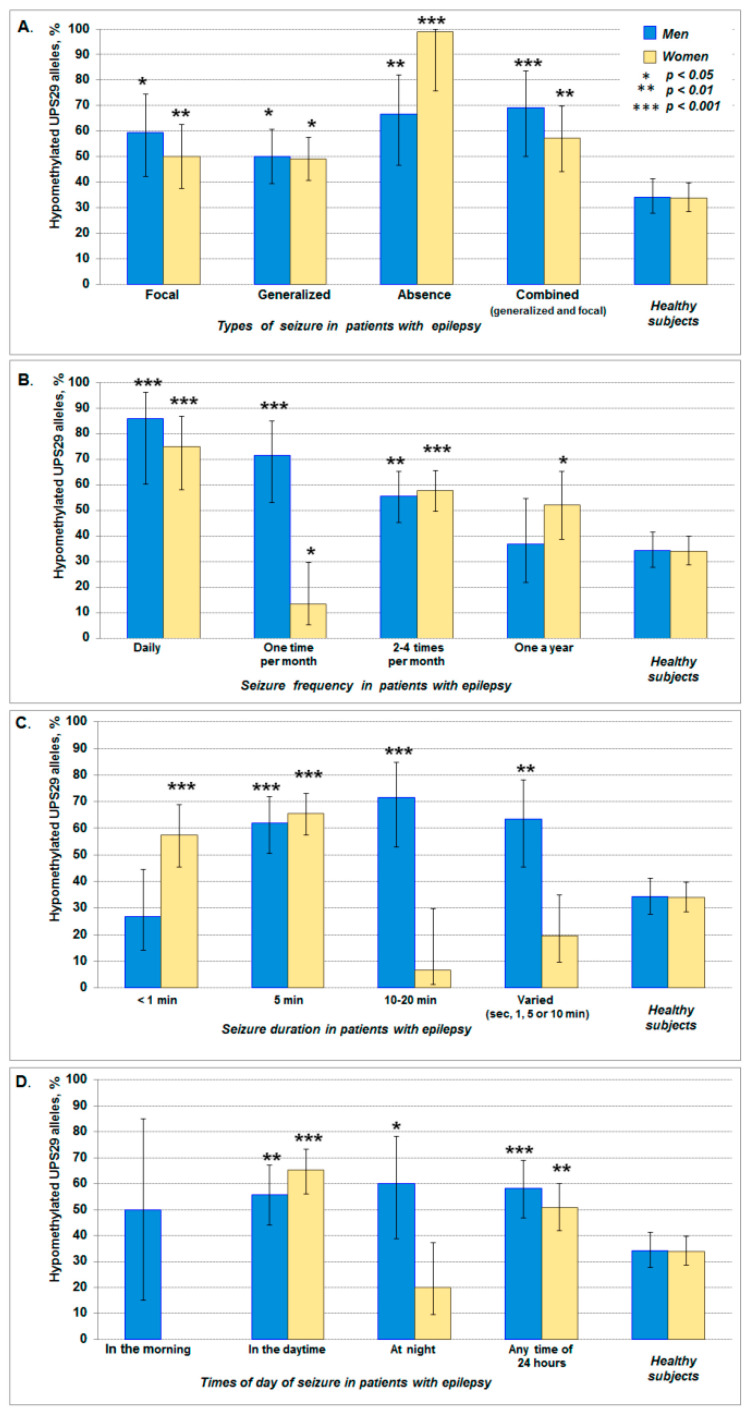
The frequency of hypomethylated (unmethylated) UPS29 alleles in patients with different clinical patterns of seizures: Type (**A**), frequency (**B**), duration (**C**), and the time of day of seizure onset (**D**). The data is shown with 95% CI. The asterisk indicates statistically significant differences in comparison with control (healthy subjects).

**Table 1 ijms-21-09206-t001:** Allele and genotype frequency of UPS29 in men with epilepsy (symptomatic and cryptogenic) compared with healthy subjects.

UPS29 Minisatellite	Symptomatic Epilepsy		Cryptogenic Epilepsy		Control	
	N	Frequency, % (95% CI)	N	Frequency, % (95% CI)	N	Frequency, % (95% CI)
**Allele**						
17 repeats	119	87.5 (80.9–92.1)	24	85.7 (68.5–94.3)	170	92.4 (87.6–95.4)
9 repeats	10	7.4 (4.0–13.0)	1	3.6 (0.6–17.7)	8	4.4 (2.2–8.4)
8 repeats	5	3.7 (1.6–8.3)	2	7.1 (2.0–22.6)	3	1.6 (0.6–4.7)
6 repeats	2	1.5 (0.4–5.2)	1	3.6 (0.6–17.7)	3	1.6 (0.6–4.7)
*Short alleles (sum)*	17	12.5 (8.0–19.1)	4	14.3 (5.7–31.5)	14	7.6 (4.6–12.4)
Total alleles	136		28		184	
*p*-value	0.181		0.269			
**Genotype**						
17/17	52	76.5 (65.1–85.0)	10	71.4 (45.4–88.3)	80	87.0 (78.6–92.4)
17/9	9	13.2 (7.1–23.3)	1	7.1 (1.3–31.5)	4	4.4 (1.7–10.7)
17/8	4	5.9 (2.3–14.2)	2	14.3 (4.0–40.0)	3	3.3 (1.1–9.1)
17/6	2	2.9 (0.8–10.1)	1	7.14 (1.3–31.5)	3	3.3 (1.1–9.1)
9/8	1	1.5 (0.3–7.9)	0	0 (0–21.5)	0	0 (0–4.0)
9/9	0	0 (0–5.4)	0	0 (0–21.5)	2	2.2 (0.6–7.6)
*Short alleles carriers (sum)*	16	23.5 (15.0–34.8)	4	28.57 (11.7–54.7)	12	13. 0 (7.6–21.4)
Sample size	68		14		92	
*p*-value	0.096		0.220			

Comment: In men from these samples we did not reveal homozygotes 8/8 and 6/6, and heterozygotes 9/6 and 8/6, and subjects with 24, 14, and 10 repeat alleles. The *p*-value is indicated for comparison with control (healthy subjects). In these samples, we compared the frequencies of short UPS29 alleles (total) and carriers of short alleles (total).

**Table 2 ijms-21-09206-t002:** Allele and genotype frequency of UPS29 in women with epilepsy (symptomatic and cryptogenic) compared with healthy subjects.

UPS29 Minisatellite	Symptomatic Epilepsy		Cryptogenic Epilepsy		Control	
	N	Frequency, % (95% CI)	N	Frequency, % (95% CI)	N	Frequency, % (95% CI)
**Allele**						
17 repeats	174	83.7 (78.0–88.1)	36	78.3 (64.4–87.7)	250	93.3 (89.6–95.7)
9 repeats	20	9.6 (6.3–14.4)	6	13.0 (6.1–25.7)	7	2.6 (1.3–5.3)
8 repeats	7	3.4 (1.6–6.8)	1	2.2 (0.4–11.3)	6	2.2 (1.0–4.8)
6 repeats	7	3.4 (1.6–6.8)	3	6.5 (2.2–17.5)	5	1.9 (0.8–4.3)
*Short alleles (sum)*	34	16.4 (11.9–22.0)	10	21.7 (12.3–35.6)	18	6.7 (4.3–10.4)
Total alleles	208		46		268	
*p*-value	<0.001		0.003			
**Genotype**						
17/17	73	70.2 (60.8–78.1)	13	56.5 (36.8–74.4)	117	87.3 (80.6–91.9)
17/9	16	15.4 (9.7–23.5)	6	26.1 (12.6–46.5)	7	5.2 (2.6–10.4)
17/8	5	4.8 (2.1–10.8)	1	4.4 (0.8–21.0)	4	3.0 (1.2–7.4)
17/6	7	6.7 (3.3–13.2)	3	13.0 (4.5–32.1)	5	3.7 (1.6–8.4)
9/8	2	1.9 (0.5–6.7)	0	0 (0–14.3)	0	0 (0–2.8)
9/9	1	1.0 (0.2–5.3)	0	0 (0–14.3)	0	0 (0–2.8)
8/8	0	0 (0–3.6)	0	0 (0–14.3)	1	0.8 (0.1–4.1)
*Short alleles carriers (sum)*	31	29.8 (21.9–39.2)	10	43.5 (25.6–63.2)	17	12.7 (8.1–19.4)
Sample size	104		23		134	
*p*-value	*0.001*		*0.001*			
*Coefficient of association, phi*	*+0.21*		*+0.29*			
*RR (95% CI)*	2.5 (1.4–4.0)		3.4 (1.9–6.5)			
*OR (95% CI)*	2.9 (1.2–5.7)		5.3 (2.0–14.0)			

Comment: In women from these samples we did not reveal homozygotes 6/6, heterozygotes 9/6, 8/6, and subjects with 24, 14, and 10 repeat alleles. The *p*-value is indicated for comparison with control (healthy subjects). In these samples, we compared the frequencies of short UPS29 alleles (total) and carriers of short alleles (total).

**Table 3 ijms-21-09206-t003:** The age at onset of epilepsy and UPS29 genotypes.

Epilepsy Type	Age at Onset of a Seizure, Years (M ± SD)					
	Men			Women		
	Excluding UPS29 Genotype	UPS29 Homozygotes 17/17	Subjects with Short UPS29 Alleles	Excluding UPS29 Genotype	UPS29 Homozygotes 17/17	Subjects with Short UPS29 Alleles
Symptomatic	19.8 ± 16.3	17.5 ± 14.5	25.5 ± 19.5	20.4 ± 17.2	21.1 ± 16.6	18.4 ± 19.5
Cryptogenic	11.9 ± 5.7	11.9 ± 6.6	11.8 ± 2.3	14.0 ± 6.3	11.6 ± 5.4	17.3 ± 6.3
Kruskal–Wallis chi-squared statistic	3.625			2.499		
*p*-value	0.305			0.475		

**Table 4 ijms-21-09206-t004:** Association of UPS29 genotype with the type of seizure at the onset.

Epilepsy Type	Seizure Types (at Onset)	Total Cases, N	Frequency of UPS29 Genotype, % (95% CI)		*p*-Value *	*phi*	RR (95% CI), OR (95% CI)
			Homozygotes 17/17	Subjects with Short UPS29 Alleles			
**Men**							
Symptomatic	Generalized	47	83.0 (69.9–91.1) n = 39	17.0 (8.9–30.1) n = 8	0.611	-	-
	Focal (partial)	21	62.0 (40.9–79.3) n = 13	38.1 (20.8–59.1) n = 8	0.012	+0.26	2.9 (1.4–6.2) 4.1 (1.4–12.0)
Cryptogenic	Generalized	11	72.7 (43.4–90.3) n = 8	27.3 (9.7–56.6) n = 3	0.359	-	-
	Focal (partial)	3	66.7 (20.8–93.9) n = 2	33.3 (6.2–79.2) n = 1	0.361	-	-
Total	Generalized	58	81.0 (69.1–89.1) n = 47	19.0 (10.9–30.9) n = 11	0.358	-	-
	Focal (partial)	24	62.5 (42.7–78.8) n = 15	37.5 (21.2–57.3) n = 9	0.009	+0.26	2.9 (1.4–6.0) 4.0 (1.4–11.2)
**Women**							
Symptomatic	Generalized	73	65.8 (54.3–75.6) n = 48	34.3 (24.4–45.7) n = 25	<0.001	+ 0.26	2.7 (1.6–4.7) 3.6 (1.8–7.2)
	Focal (partial)	31	80.7 (63.7–90.8) n = 25	19.4 (9.18–36.3) n = 6	0.387	-	-
Cryptogenic	Generalized	15	53.3 (30.1–75.2) n = 8	46.7 (24.8–69.9) n = 7	0.003	+ 0.28	3.7 (1.8–7.4) 6.0 (1.9–18.7)
	Focal (partial)	8	62.5 (30.6–86.3) n = 5	37.5 (13.7–69.4) n = 3	0.085	-	-
Total	Generalized	88	63.6 (53.2–72.9) n = 56	36.4 (27.1–46.8) n = 32	<0.001	+ 0.28	2.9 (1.7–4.8) 3.9 (2.0–7.7)
	Focal (partial)	39	76.9 (61.7–87.4) n = 30	23.1 (12.7–38.3) n = 9	0.128	-	-

Comment: * a comparison with control, *phi*-coefficient of association, n-the number of cases.

**Table 5 ijms-21-09206-t005:** The frequency of seizure types at the onset of epilepsy in men and women.

Epilepsy Type	Seizure Types (at Onset)	Men		Women	
		N	Frequency, % (95% CI)	N	Frequency, % (95% CI)
Symptomatic	Generalized	47	69.1 (57.4–78.8)	73	70.2 (60.8–78.1)
	Focal (partial)	21	30.9 (21.2–42.6)	31	29.8 (21.9–39.2)
Cryptogenic	Generalized	11	78.6 (52.4–92.4)	15	65.2 (44.9–81.2)
	Focal (partial)	3	21.4 (7.6–47.6)	8	34.8 (18.8–55.1)
Total	Generalized	58	70.7 (60.1–79.5)	88	69.3 (60.8–76.7)
	Focal (partial)	24	29.3 (20.5–39.9)	39	30.7 (23.4–39.2)

**Table 6 ijms-21-09206-t006:** Etiological factors, structural changes in the brain, and neurological status in patients with epilepsy.

Sample	Frequency, % (95% CI)	
	Men	Women
***Etiological factors***		
Perinatal pathology	41.2 (30.3–53.0)	55.8 (46.2–65.0)
Traumatic brain injuries	35.3 (25.0–47.2)	24.0 (16.9–33.1)
Cerebrovascular pathologies	10.3 (5.1–19.8)	10.6 (6.0–18.0)
Neuroinfection and intoxication	11.8 (6.1–21.5)	6.7 (3.3–13.2)
Brain tumors	1.5 (0.3–7.9)	2.9 (1.0–8.1)
Total, N	68	104
***Structural changes in the brain ****		
Hydrocephalus (and arachnoid cysts)	57.4 (45.5–68.4)	47.1 (37.8–56.6)
Cortical atrophy	32.4 (22.4–44.2)	26.0 (18.5–35.1)
Brain tumors	10.3 (5.1–19.8)	26.9 (19.3–36.2)
Total, N	68	104
***Neurological status***		
Autonomic dysfunction	34.3 (24.1–46.3)	21.1 (14.1–30.3)
Ataxia	22.4 (14.1–33.7)	21.1 (14.1–30.3)
Cranial nerve impairment	14.9 (8.3–25.4)	15.8 (9.8–24.4)
Pyramidal system disturbances	13.4 (7.2–23.6)	19.0 (7.2–23.6)
Extrapyramidal system disorders	7.5 (3.2–16.3)	9.5 (5.1–17.0)
Sensory impairment	7.5 (3.2–16.3)	13.7 (8.2–22.0)
Total, N	67	95

Comment: * *p*-value = 0.029 (comparison of men with women).

**Table 7 ijms-21-09206-t007:** Clinical characteristics of seizures in patients with epilepsy.

Sample	Frequency, % (95% CI)	
	Men	Women
**Seizure types**		
Focal (partial)	19.5 (12.4–29.4)	22.8 (16.4–30.9)
Generalized	50.0 (39.4–60.6)	51.2 (42.6–59.7)
Absence	14.6 (8.6–23.9)	4.7 (2.2–9.9)
Combined	15.9 (9.5–25.3)	21.3 (15.0–29.2)
**Seizure frequency**		
Daily	8.5 (4.0–16.6)	12.6 (7.9–19.5)
One time per month	17.1 (10.5–26.6)	11.8 (7.3–18.6)
2–4 times per month	56.1 (45.3–66.3)	55.9 (47.2–64.3)
Once a year	18.3 (11.4–28.0)	19.7 (13.7–27.5)
**Seizure duration ***		
<1 min	18.3 (11.4–28.0)	26.0 (19.1–34.2)
5 min	46.3 (36.0–57.1)	54.3 (45.7–62.7)
10–20 min	17.1 (10.5–26.6)	5.5 (2.7–10.9)
Varied (sec, 1, 5 or 10 min)	18.3 (11.4–28.0)	14.2 (9.2–21.3)
**Times of day for seizure onset**		
In the morning	2.4 (0.7–8.5)	0 (0.0–2.9)
In the daytime	41.5 (31.4–52.3)	44.1 (35.8–52.8)
At night	12.2 (6.8–21.0)	11.8 (7.3–18.6)
Any time of 24 h	43.9 (33.7–54.7)	44.1 (35.8–52.8)
Total, N	82	127

Comment: * *p*-value = 0.028 (comparison of men with women).

**Table 8 ijms-21-09206-t008:** The frequency of methylated and hypomethylated (unmethylated) UPS29 alleles in patients with epilepsy (symptomatic and cryptogenic) compared with the control group.

UPS29 Epialleles	Symptomatic Epilepsy			Cryptogenic Epilepsy			Control	
	N	Frequency, % (95% CI)	*p*-Value	N	Frequency, % (95% CI)	*p*-Value	N	Frequency, % (95% CI)
**Men**								
L^meth+^	49	36.0 (28.5–44.4)	<0.001	12	42.9 (26.5–60.9)	0.170	113	61.4 (54.2–68.1)
L^meth−^	70	51.5 (43.2–59.7)		12	42.9 (26.5–60.9)		57	31.0 (24.7–38.0)
Sh^meth+^	7	5.2 (2.5–10.3)	0.479	2	7.1 (2.0–22.6)	1.000	8	4.4 (2.2–8.4)
Sh^meth−^	10	7.4 (4.0–13.0)		2	7.1 (2.0–22.6)		6	3.3 (1.5–6.9)
*Methylated alleles (sum)*	56	41.2 (33.3–49.6)	<0.001	14	50.0 (32.6–67.4)	0.139	121	65.8 (58.6–72.2)
*Hypomethylated alleles (sum)*	80	58.8 (50.4–66.7)		14	50.0 (32.6–67.4)		63	34.2 (27.8–41.4)
Total	136			28			184	
*Coefficient of association, phi*	+0.24			-				
*RR (95% CI)*	1.6 (1.3–2.0)			-				
*OR (95% CI)*	2.7 (1.7–4.3)			-				
**Women**								
L^meth+^	85	40.9 (34.4–47.7)	<0.001	14	30.4 (19.1–44.8)	0.002	165	61.5 (55.6–67.29)
L^meth−^	89	42.8 (36.3–49.6)		22	47.8 (34.1–61.9)		85	31.7 (26.4–37.5)
Sh^meth+^	14	6.7 (4.1–11.0)	0.144	5	10.9 (4.7–23.0)	0.444	12	4.5 (2.68–7.7)
Sh^meth−^	20	9.6 (6.3–14.4)		5	10.9 (4.7–23.0)		6	2.3 (1.0–4.8)
*Methylated alleles (sum)*	99	47.6 (40.9–54.4)	<0.001	19	41.3 (28.3–55.7)	0.002	177	66.0 (60.2–71.5)
*Hypomethylated alleles (sum)*	109	52.4 (45.6–59.1)		27	58.7 (44.3–71.7)		91	34.0 (28.6–39.8)
Total	208			46			268	
*Coefficient of association, phi*	+0.19			+0.18				
*RR (95% CI)*	1.4 (1.2–1.6)			1.6 (1.1–2.3)				
*OR (95% CI)*	2.1 (1.5–3.1)			2.8 (1.5–5.2)				

Comment: L^meth+^-methylated long UPS29 allele (17 repeats); L^meth−^- hypomethylated (unmethylated) long UPS29 allele (17 repeats); Sh^meth+^-methylated short UPS29 alleles (total 9, 8, and 6 repeats); Sh^meth−^-hypomethylated (unmethylated) short UPS29 alleles (total 9, 8, and 6 repeats); and *p*-value is indicated for a comparison with control (healthy subjects).

**Table 9 ijms-21-09206-t009:** The frequency of UPS29 epigenotypes in patients with epilepsy (symptomatic and cryptogenic) compared with the control group.

UPS29 Epigenotypes	Symptomatic Epilepsy		Cryptogenic Epilepsy		Control	
	N	Frequency, % (95% CI)	N	Frequency, % (95% CI)	N	Frequency, % (95% CI)
**Men**						
L^meth+^/L^meth+^	23	33.8 (23.7–45.7)	5	35.7 (16.3–61.2)	54	58.7 (48.5–68.2)
L^meth−^/L^meth−^	29	42.7 (31.6–54.5)	5	35.7 (16.3–61.2)	26	28.3 (20.1–38.2)
L^meth+^/Sh^meth+^	3	4.4 (1.5–12.2)	2	14.3 (4.0–40.0)	5	5.4 (2.3–12.1)
L^meth+^/Sh^meth−^	0	0 (0–3.4)	0	0 (0–21.5)	0	0 (0–4.0)
L^meth−^/Sh^meth+^	2	2.9 (0.8–10.1)	0	0 (0–21.5)	1	1.1 (0.2–5.9)
L^meth−^/Sh^meth−^	10	14.7 (8.2–25.0)	2	14.3 (4.0–40.0)	4	4.3 (1.7–10.7)
Sh^meth+^/Sh^meth+^	1	1.5 (0.3–7.9)	0	0 (0–21.5)	1	1.1 (0.2–5.9)
Sh^meth+^/Sh^meth−^	0	0 (0–3.4)	0	0 (0–21.5)	0	0 (0–4.0)
Sh^meth−^/Sh^meth−^	0	0 (0–3.4)	0	0 (0–21.5)	1	1.1 (0.2–5.9)
*Subjects with methylated UPS29* *(sum)*	29	42.6 (31.6–54.5)	7	50.0 (26.8–73.2)	61	66.3 (56.2–75.1)
*Subjects with hypomethylated UPS29 (sum)*	39	57.4 (45.5–68.4)	7	50.0 (26.8–73.2)	31	33.7 (24.9–43.8)
Sample size	68		14		92	
*p-value*	0.004		0.370			
*Coefficient of association, phi*	+0.24		-			
*RR (95% CI)*	1.6 (1.1–2.1)		-			
*OR (95% CI)*	2.7 (1.4–5.1)		-			
**Women**						
L^meth+^/L^meth+^	38	36.5 (27.9–46.1)	5	21.7 (9.7–41.9)	80	59.7 (51.2–67.6)
L^meth−^/L^meth−^	35	33.7 (25.3–43.2)	8	34.8 (18.8–55.1)	37	27.6 (20.7–35.7)
L^meth+^/Sh^meth+^	7	6.7 (3.3–13.2)	3	13.0 (4.5–32.1)	4	3.0 (1.2–7.4)
L^meth+^/Sh^meth−^	2	1.9 (0.5–6.7)	1	4.3 (0.8–21.0)	1	0.7 (0.1–4.1)
L^meth−^/Sh^meth+^	2	1.9 (0.5–6.7)	2	8.7 (2.4–26.8)	6	4.5 (2.1–9.4)
L^meth−^/Sh^meth−^	17	16.3 (10.5–24.6)	4	17.4 (7.0–37.1)	5	3.7 (1.6–8.4)
Sh^meth+^/Sh^meth+^	2	1.9 (0.5–6.7)	0	0 (0–14.3)	1	0.7 (0.1–4.1)
Sh^meth+^/Sh^meth−^	1	1.0 (0.2–5.2)	0	0 (0–14.3)	0	0 (0–2.8)
Sh^meth−^/Sh^meth−^	0	0 (0–3.6)	0	0 (0–14.3)	0	0 (0–2.8)
*Subjects with methylated UPS29* *(sum)*	52	50.0 (40.6–59.4)	11	47.8 (29.2–67.0)	92	68.7 (60.4–75.9)
*Subjects with hypomethylated UPS29 (sum)*	52	50.0 (40.6–59.4)	12	52.2 (33.0–70.8)	42	31.3 (24.1–39.6)
Sample size	104		23		134	
*p*-value	0.005		0.060			
*Coefficient of association, phi*	+0.19		-			
*RR (95% CI)*	1.4 (1.1–1.7)		-			
*OR (95% CI)*	2.2 (1.3–3.7)		-			

Comment: L^meth+^-methylated long UPS29 allele (17 repeats); L^meth−^-hypomethylated (unmethylated) long UPS29 allele (17 repeats); Sh^meth+^-methylated short UPS29 alleles (total 9, 8, and 6 repeats); Sh^meth−^-hypomethylated (unmethylated) short UPS29 alleles (total 9, 8, and 6 repeats); and *p*-value is indicated for a comparison with control (healthy subjects).

**Table 10 ijms-21-09206-t010:** The association of UPS29 hypomethylation with a clinical pattern of seizures in patients with epilepsy.

Sample	Men				Women			
	*p*-Value	RR (95% CI)	OR (95% CI)	*phi*	*p*-Value	RR (95% CI)	OR (95% CI)	*phi*
**Seizure types**								
Focal (partial)	0.020	1.5 (1.1–2.0)	1.9 (1.1–3.3)	+0.15	0.004	1.5 (1.1–1.9)	1.9 (1.2–2.9)	+0.15
Generalized	0.010	1.7 (1.2–2.5)	2.8 (1.3–6.1)	+0.18	0.025	1.5 (1.1–2.0)	2.0 (1.1–3.5)	+0.13
Absence	0.003	2.0 (1.4–2.8)	3.8 (1.6–9.5)	+0.21	<0.001	3.0 (2.9–3.5)	23.3 (3.0–18.2)	+0.28
Combined	0.001	2.0 (1.5–2.8)	4.3 (1.8–10.5)	+0.24	0.001	1.7 (1.3–2.3)	2.6 (1.5–4.8)	+0.18
**Seizure frequency**								
Daily	< 0.001	2.5 (1.9–3.4)	11.5 (2.5–53.1)	+0.27	<0.001	2.2 (1.7–2.9)	5.8 (2.5–13.5)	+0.26
One time per month	< 0.001	2.1 (1.5–2.8)	4.8 (2.0–11.5)	+0.26	0.023	2.6 (1.0–6.4)	3.3 (1.1–9.9)	−0.13
2–4 times per month	0.001	1.6 (1.2–2.1)	2.4 (1.4–4.0)	+0.20	<0.001	1.7 (1.4–2.1)	2.7 (1.8–4.0)	+0.23
Once a year	0.837	-	-	-	0.017	1.5 (1.1–2.1)	2.1 (1.2–3.9)	+0.14
**Seizure duration**								
< 1 min	0.532	-	-	-	0.001	1.7 (1.3–2.2)	2.6 (1.5–4.6)	+0.19
5 min	< 0.001	1.8 (1.4–2.4)	3.1 (1.8–5.4)	+0.25	<0.001	2.0 (1.6–2.4)	3.7 (2.4–5.8)	+0.30
10–20 min	< 0.001	2.1 (1.5–2.8)	4.8 (2.0–11.5)	+0.26	0.042	-	-	-
Varied (sec, 1, 5 or 10 min)	0.003	1.9 (1.3–2.6)	3.3 (1.5–7.4)	+0.21	0.090	-	-	-
**Times of day for seizure onset**								
In the morning	0.610	-	-	-	-	-	-	-
In the daytime	0.002	1.6 (1.2–2.2)	2.4 (1.4–4.3)	+0.20	<0.001	1.9 (1.6–2.4)	3.6 (2.3–5.8)	+0.29
At night	0.028	1.8 (1.2–2.6)	2.9 (1.2–7.4)	+0.16	0.152	-	-	-
Any time of 24 h	0.001	1.7 (1.3–2.3)	2.7 (1.5–4.7)	+0.22	0.003	1.5 (1.2–2.0)	2.0 (1.3–3.2)	+0.16

Comment: *p*-value is indicated for comparison with control (healthy subjects).

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
