# Peer review of "Length Polymorphism and Methylation Status of UPS29 Minisatellite of the ACAP3 Gene as Molecular Biomarker of Epilepsy. Sex Differences in Seizure Types and Symptoms"

_ijms, 2020, doi:10.3390/ijms21239206_

Round 1

Reviewer 1 Report

The authors have addressed my concerns on the inconsistency of the data in the text and the table.  But, there are still numerous errors in English writing.  

Author Response

All corrections are shown in Word in edit mode as notes. All corrections are shown by violet color in the text.

Reviewer 2 Report

The explanations are quite satusfactory and I think improved the manuscript significantly. 

Author Response

All notes were related to English.

All corrections are shown in Word in Edit mode as notes. All corrections are shown by violet color in the text.

This manuscript is a resubmission of an earlier submission. The following is a list of the peer review reports and author responses from that submission.

Round 1

Reviewer 1 Report

This retrospective study investigated the length polymorphism and DNA methylation of UPS29 minisatellite of ACAP3gene in patients with epilepsy.  They observed that both genetic (UPS29 minisatellite polymorphism) and epigenetic (DNA methylation) factors contribute to epilepsy development and clinical patterns of seizures in a sex-dependent manner. 

However, the data were poorly presented with some inconsistent description (e.g., lines 212-215, lines 223-225) and missing datasets (e.g., Tables 5 and 6), and the Results section is hard to follow.  Also, the manuscript is not well organized; a subsection of Conclusion was included in the Discussion section, but a separate Conclusion section was added to the end of the manuscript (prior to References). In addition, there are numerous grammatical errors, typos and formatting issues; a list of examples is as follows: Lines 28-30, lines 53-55, line 63, line 118, lines 284-288, line 472, line 514, line 532 and line 899.

Other issues for consideration:

  • Abbreviations should be defined when they first appear in the text, such as VNTR, SNPs.
  • Consistent symbols should be used to represent the same terms, such as 17/17 for 17 repeats homozygotes.
  • Why 1989 but not 2017 ILAE classification of epilepsies and epileptic syndromes was used in this study?

Author Response

Responses to review 1.

  1. Reviewer. The data were poorly presented with some inconsistent description (e.g., lines 212-215, lines 223-225) and missing datasets (e.g., Tables 5 and 6), and the Results section is hard to follow.

Response. Necessary data were added to table 4. Also, two tables 5 and 6 were added in manuscript. Thus results become more clear.

  1. Reviewer. Also, the manuscript is not well organized; a subsection of Conclusion was included in the Discussion section, but a separate Conclusion section was added to the end of the manuscript (before References).

Response. All was corrected.

  1. Reviewer. there are numerous grammatical errors, typos, and formatting issues; a list of examples is as follows: Lines 28-30, lines 53-55, line 63, line 118, lines 284-288, line 472, line 514, line 532, and line 899.

Response. All these and other errors, typos, and formatting were corrected.

  1. Reviewer. •Abbreviations should be defined when they first appear in the text, such as VNTR, SNPs.
  • Consistent symbols should be used to represent the same terms, such as 17/17 for 17 repeats homozygotes.

Response. All abbreviations were defined in the text and an additional list of abbreviations is added at the end after references. Symbols 17/17 are corrected in Table 2.

  1. Reviewer • Why 1989 but not 2017 ILAE classification of epilepsies and epileptic syndromes was used in this study?

Response.       Our studied have been begun in the 2008 year when the classification of epilepsy and epileptic syndromes was based on the 1989 classification during the analysis of associations of UPS29 with epilepsy. Later we conducted a more deep analysis and used a new classification of the 2017 year. Thus according to the advice of the referee we added the information into Materials and Methods.

Reviewer 2 Report

This research work is good one to explain different epileptic phenotypes associated with repeat polymorphism and methylation status of UPS29 gene. To me this one is an interesting study but with some loopholes. The specific comments are as follows:-

  1. The sample number for association analysis is too small for some cases. As for example in Table 1 total allele 136, but number of 6 repeats is only 2, which is too small. I suggest total allele should increase 250 to 300, so that all allele types have N that are ≥
  2. Also there is significant difference in total number of alleles (N) between Table 1 vs Table 2 (136 vs 208). I suggest they should be comparable. Otherwise difference in disease association between men and women become illusive.
  3. Why long alleles are just one type (17 repeats), authors should incorporate 24 or 10 repeat allele frequency. Or if these alleles are absent in the given population authors should mention it clearly with proper references.
  4. Authors did not find any significant association between short allele frequency and Epilepsy among men, but they showed significant association between etiological factors, structural changes, or neurological conditions among male epileptic patients  with short allele frequency compared to healthy substances. But this can not be explained in biological point of view. Authors should explain the probable  reasons for this discrepancy.
  5. It will be interesting to see whether there is association between specific genotype( eg. homozygous short allele vs  heterozygous short allele) with phenotypic or cryptogenic epilepsy.
  6. Why homozygous hypomethylated short allele is absent ? Do authors have any explanation?
  7. To explain molecular mechanism behind this disease association with hypomethylated short allele of UPS29 would be stronger if author could present mRNA/protein level of ACAP3 in patients harboring hypomethylated short allele vs hyper methylated long alleles.

Author Response

Responses  to review 2.

1.Reviewer.    The sample number for association analysis is too small for some cases. For example in Table 1 total allele 136, but the number of 6 repeats is only 2, which is too small. I suggest total allele should increase 250 to 300 so that all allele types have N that are ≥

Response.  Indeed, at first glance, the sample is rather small but it is enough for nonparametric statistical criteria. Our earlier data [ref. 31, 60] and later studies (unpublished data) on 450 healthy volunteers showed that short alleles frequency in the Caucasian population, is low and the size of the sample does not influence observed allele frequency.

  1. Reviewer. Also, there is a significant difference in the total number of alleles (N) between Table 1 vs Table 2 (136 vs 208). I suggest they should be comparable. Otherwise, the difference in disease association between men and women becomes illusive.

Response.  The difference in the total number of alleles between men and women is explained by more women with epilepsy which get to the clinic than men.  Artificial aligning samples will lead to technical distortion of the real frequency of UPS 29 alleles in patients.  Besides, used statistical criteria permit to compare samples of different sizes.

  1. Reviewer. Why long alleles are just one type (17 repeats), authors should incorporate 24 or 10 repeat allele frequencies. Or if these alleles are absent in the given population authors should mention it clearly with proper references.

Response. In this work in studied samples (below 50 years old), there were not found alleles consisted of 24, 14, and 10 repeats. This information was added to the text. I t is necessary to note that such alleles were found by us earlier in a group of patients over 70 years old.

  1. Reviewer. Authors did not find any significant association between short allele frequency and Epilepsy among men, but they showed a significant association between etiological factors, structural changes, or neurological conditions among male epileptic patients with short allele frequency compared to healthy substances. But this can not be explained from a biological point of view. Authors should explain the probable reasons for this discrepancy.

Response. This discrepancy just underlines the imperfection of the classification of epilepsy into symptomatic, cryptogenic, and idiopathic, and speaks in favor of the classification of epilepsy according to the type of seizures at the onset of the disease and during the disease, as well as by specific etiological factors of the disease (where this can be established during clinical examination and patient interview)

  1. Reviewer. It will be interesting to see whether there is an association between specific genotype( eg. homozygous short allele vs heterozygous short allele) with phenotypic or cryptogenic epilepsy.

Response. A very good point. We also wanted to test whether carriers of short UPS29 alleles have a correlation between a certain genotype and the clinical characteristics of seizures, etc. But to our great regret, the frequency of occurrence of short alleles among Caucasians (and especially in the homozygous state) is low, which does not allow for an adequate statistical analysis with such a variant of the formation of the analyzed (compared) groups (namely, to compare homozygote and heterozygotes for short alleles UPS29)

  1. Reviewer. Why homozygous hypomethylated short allele is absent? Do authors have any explanation?

Response. For the same reason as described above (among Caucasians, short UPS29 alleles in a homozygous state are very rare). Therefore, in this work, we were unable to identify hypomethylated short alleles in the homozygous state.

  1. Reviewer. To explain the molecular mechanism behind this disease associated with the hypomethylated short allele of UPS29 would be stronger if the author could present the mRNA/protein level of ACAP3 in patients harboring hypomethylated short allele vs hypermethylated long alleles.

Response. Again for the same reason: Among Caucasians, short UPS29 alleles in a homozygous state of occurrence are very rare, therefore it was not possible to conduct an adequate statistical analysis of the mRNA / protein level of ACAP3 expression in patients with epilepsy, in whom this allele is in a hypomethylated state in comparison with similar genotype, but in a methylated state, as well as in comparison with similar genotypes / epigenotypes in the control group in men and women.

Round 2

Reviewer 1 Report

The authors have addressed some of my concerns.  However, there are still numerous grammatical errors, typos and formatting issues.  Significant improvement in language is required to reach publication standard.  Also, the Results part is still hard to follow.  For example, from line 221 to line 241, were those numbers cited from Table 4 and Table 5?  If so, why don’t they match those in tables?  

Author Response

Dear reviewer,

First of all I would like to thank you very much for such careful reading of manuscript. All your remarks are very important to improve the content and understanding of results. Below there are responses to your remarks.

Response.

Reviewer. : the inconsistency of the data presented when I began to compare the% figures in the table and in the text.

Reply. This discrepancy is explained by the fact that Table 4 shows the frequencies of "genotypes" (Homozygotes 17/17 and Subjects with short UPS29 alleles) calculated within the "Seizure types at onset" groups (i.e.,% of the total number of patients in group with generalized seizures and similarly from the total number of patients in the group with focal seizures). This was done for comparison with male and female controls. Whereas in the text (lines from 221 to 241, in the new version of the manuscript) the frequency of occurrence of the type of seizure (generalized or focal) at the onset of the disease is indicated, calculated already within the "Genotype" groups (that is, the frequency% of the total number of patients in the group “Subjects with short UPS29 alleles”, and similarly in the group “Homozygotes 17/17”. Whereas in the text (lines from 221 to 241, in the new version of the manuscript) the frequency of occurrence of the type of seizure (generalized or focal) at the onset of the disease is indicated, calculated already within the "Genotype" groups (that is, the frequency% of the total number of patients in the group “Subjects with short UPS29 alleles”, and similarly in the group “Homozygotes 17/17”. Therefore, for clarification, I added (highlighted in green) the values of the number of cases (n) and the total number of observations (N) in the corresponding "genotypes" - "types of seizures" to table 4 and in the text.

          In this regard, perhaps, it is necessary to comment the table â„–5.

It indicates the frequency of occurrence of generalized and focal seizures in general in the population (for men and women) without taking into account the UPS29 genotype, therefore the numbers (%, frequency of cases) do not coincide in this table. from Table 4 and texts. These results complement each other, and it seems to me that they emphasize the existence of a connection between the UPS29 genotype and the type of seizures at the onset of epilepsy (in particular, sex differences in seizure types).

  1. Second remark concerns English. Indeed we again found errors, and all were corrected as much as possible for non-native speakers.